# HIV risk behavior and HIV testing among rural and urban men who have sex with men in Zhejiang Province, China: A respondent-driven sampling study

**Lin He[1], Xiaohong Pan** [1]*, **Jiezhe Yang[1], Qiaoqin Ma[1]***, **Jun Jiang[1], Wei Wang[2], Jiaquan Qiu[3], Yazhou Zou[2], Ping Wang[4], Dongshe Zhao[5], Hui Wang[1], Tingting Jiang[1]**

**1** Zhejiang Provincial Center for Disease Control and Prevention, Hangzhou, Zhejiang, China, **2** Quzhou Center for Disease Control and Prevention, Quzhou, Zhejiang, China, **3** Kaihua county Center for Disease Control and Prevention, Quzhou, Zhejiang, China, **4** Jiangshan Center for Disease Control and Prevention, Quzhou, Zhejiang, China, **5** Lucheng district Center for Disease Control and Prevention, Quzhou, Zhejiang, China

* xhpan@cdc.zj.cn (XP); qqma@cdc.zj.cn (QM)

**Data Availability Statement:** Data cannot be shared publicly because of some of participants are HIV-infected patients. Data are available from the Institutional Review Board of the Zhejiang

## Abstract

### Background

Currently, human immunodeficiency virus (HIV) sentinel surveillance among men who have sex with men (MSM) in China conducted in large and medium-sized cities, and no HIV sentinel surveillance conducted in rural areas. HIV testing and intervention is predominantly conducted in urban areas, there have been a limited number of studies in rural areas MSM, it is necessary to conduct the investigation of HIV risk sexual behavior, HIV testing among rural and urban MSM.

### Method

Between December 2013 and August 2015, a cross-sectional study was conducted in rural and urban areas in Zhejiang Province using respondent-driven sampling (RDS). Participants completed face-to-face interviewer-administered questionnaire surveys and were tested for HIV.

### Results

A total of 710 MSM participants were recruited, of whom 36.1% were from rural areas. The overall HIV prevalence was 16.6%, and was considerably lower among MSM living in rural areas (3.9%) than those living in urban areas (24.2%). 61.1% participants had not condom use with male sexual behavior in the past 6 months (86.7% in rural areas and 46.7% in urban areas). The social demographic and behavioral characteristics had significance difference among rural and urban MSM. Multivariate logistic regression revealed that, compared to men living in urban areas, MSM living in rural areas MSM were more likely to use dating apps to find sexual partners, were more likely to engage in bisexual behavior, and had lower condom use. 43.0% participants had been tested for HIV in the past year (41.8% in rural

Provincial Center for Disease Control and Prevention for researchers who meet the criteria for access to confidential data. You can contact Zhenggang Jiang, who is the director of ethics committee, the email is zhgjiang@cdc.zj.cn.

**Funding:** The study was funded by a grant from 'National S & T Major Project Foundation of China' (2012ZX10001-001) to XP and (2017ZX10201101002) to XP, Zhejiang Provincial Medicine Science and Technology Plan (2015PYA004) to XP and Social Development Project of Public Welfare Technology Research in Zhejiang Province (LGF18H260006) to LH. The funding body had no role in the design of the study and collection, analysis, and interpretation of data and in writing the manuscript.

**Competing interests:** The authors have declared that no competing interests exist.

**Abbreviations:** MSM, who have sex with men; χ2, chi squared test; AOR, adjusted odds ratio; CI, confidence interval; sexually, transmitted disease, STI.

areas and 43.6% in urban areas). Multivariate logistic regression also revealed that among participants living in rural areas, having rural health insurance and not accepting HIV intervention were associated with lower HIV testing rates, while a higher monthly income and through use of internet to find sexual partner were associated with higher rates of HIV testing.

## Conclusion

High risk behavior was prevalent, and HIV testing rates were low among MSM living in rural areas compare to urban areas in Zhejiang Province, therefore, preventative intervention measures should be immediately among rural MSM urgently to reduce HIV transmission and to promote HIV testing.

## Background

Men who have sex with men (MSM) were generally a high-risk population for human immunodeficiency virus (HIV) infection because of their tendency to have multiple sexual partners and unprotected anal intercourse[1]. The proportion of HIV infections transmitted through homosexual sex increased from 12% in 2010 to 28.2% in 2018[2], and the sentinel surveillance of MSM showed the HIV prevalence increased from 5.7% to 6.9% over this period[2]. Currently, HIV intervention among MSM is predominantly conducted in cities because of the greater availability of venues such as parties, baths, and parks where men have sex with men. In 2013–2017, the HIV testing rate among MSM in China was 55.2%[3], with Beijing and Shenzhen reporting HIV testing rates of 69%[4] and 58.2%[5], respectively. However, with a lack of HIV intervention measures, rural areas (defined as people lived in towns and rural areas in county level) may have lower HIV testing rates. National estimates indicate that of the sexually active men between the ages of 15 and 49 years, 2–4% have sex with men. As half of the population of China lives in rural areas, there are a large number of MSM in living in rural areas. Therefore, in order for China is meet the United Nations Program on AIDS (UNAIDS) target of 90% of HIV infections by 2020, HIV testing of MSM living in rural areas is critical.

In China, rural areas are characterized by distance from cities and a lack of public meeting spaces, such as baths and parks. Thus, rural areas can appear to have a low MSM population density. However, a study conducted in Beijing of MSM migrant workers from rural areas who were MSM found that they were at high-risk of acquiring HIV infection[6]. In addition, a study conducted in the United States found that over 50% of MSM living in rural areas engaged in moderately high to high risk behaviors[7]. Another study found that the transmission of HIV often crossed the rural-urban divide[8]. A South African study found rural origin to be independently associated with HIV infection[9]. One study found that sexual behavior did not differ significantly between rural and urban MSM[10]. Yet, a study conducted in Yunnan, China indicated that there was a lack of resources, MSM were at risk for HIV infection [11].

In China, cultural and familial pressure drive a greater proportion of MSM to have female partners than anywhere else in the world[11, 12]. Since heterosexual marriage often conceals the orientation of MSM; therefore, HIV infection is often prevented due to monogamy. Although homosexuality is becoming more tolerable in Chinese society, high levels of stigmatization remain[13, 14]. As a result of this social stigmatization, MSM often fear social sanction

if they were to reveal their sexual orientation and continue to face the same pressures from the culturally normative social duties of heterosexual men[15]. However, this stigmatization may contribute to an increase in the rate of transmission of HIV and other sexually transmitted infections (STIs)[12].

Zhejiang Province is located on the east coast of China, is economically developed, and has had an HIV prevalence of 8% among MSM since 2008[16]. During this same time period, the national HIV prevalence was 4.9%[17]. The HIV epidemic in the Zhejiang Province forecasts a future trend for China. In China and Zhejiang province, the HIV sentinel surveillance for MSM set in urbans, and no sentinel surveillance conducted in rural areas, very little data of HIV from rural MSM. The rural population has the characteristics of short-term and/or long-term migrant to the urban areas. In addition, the rural MSM gathering in the urban places such as bathes or parks to find sexual partners, as well as the use of MSM Group data software, may promote the interaction between the urban and rural MSM. Lastly, intervention and testing for HIV is predominantly conducted in urban areas and there have been a limited number of studies on HIV-related sexual behavior, social networks, HIV prevalence, and testing information among MSM in rural areas. Very few studies have focused on MSM from rural areas, their HIV-related sexual behavior, social networks, HIV prevalence, and testing history. Identifying HIV-infected individuals living in rural areas is the key to having 90% of infections diagnosed by 2020. Therefore, this study aimed to describe HIV testing coverage and to determine factors associated with HIV testing uptake among MSM in a rural area in China.

## Methods

### Sampling and sample size calculation

Using respondent-driven sampling (RDS) to recruit participants, the study initially recruited eight participants in rural areas and five participants in urban areas as seeds. Participants were recruited from the local MSM population based on diverse characteristics, including age and other sociodemographic characteristics. The initial seeds were asked to recruit partners or peers to participate in the survey. After the seeds completed the study questionnaires and underwent HIV testing, they were provided with three recruitment coupons to incentivize other MSM to enroll in the study. Using these coupons, the seed participants then recruited a maximum of three MSM peers from their social network to participate in the survey. These new recruits were then asked to recruit up to three additional peers to participate in the study. The sample size based on the use of HIV prevalence ranged from 5% to 10%; and HIV testing in past year range from 40% to 60%. P-values≤0.05 and β = 0.1 were considered statistically significant, and precision = 0.1. Thus, it was estimated that the minimum sample size required for this study would be 402 people. The sample size was calculated using Power Analysis and Sample Size (PASS) software Version 11.0 (NCSS, LLC. Kaysville, Utah, USA).

### Study setting

We recruited two eligible rural participants as seeds who were willing to participant the survey, they were from Quzhou city, one was from Kaihua and the other from Jiangshan. For the urban we recruited eight participants as seeds who were from Lucheng District, in Wenzhou city. Kaihua and Jiangshan are rural areas located approximately 40 kilometers away from the nearest city of Quzhou. Both counties are in the southwest region of the Zhejiang Province and are characterized by hilly terrain, inconvenient traffic routes, are economic developing areas, resulting in many young adults going to urban areas for work, and there are no public meeting places for MSM, such as baths or parks. In the year, prior to the study survey, the number of MSM tested for HIV was 50 and 40 in Kaihua and Jiangshan, respectively. Lucheng

District an economically developed urban area and the unique downtown located in a coastal city in Wenzhou. Many migrant MSM work in Lucheng which has several public places such as baths and parks where MSM can meet. Quzhou and Wenzhou are representative of rural and urban areas in Zhejiang Province. In Quzhou, 44.3% residents are peasants, in contrary, 90.5% the residents in Lucheng district are citizens. The study was conducted in urban areas in Lucheng District between December 2013 and June 2014, and in rural areas between September 2014 and August 2015.

## Study participants

For this study, MSM participants were recruited based on the following inclusion criteria: (1) a minimum age of 14-yearss; (2) reported having had anal sex with men within the previous; (3) had lived in the recruited areas more than 3 months.

## Data collection

After signing informed consent, participants completed a questionnaire administered, face-to-face questionnaire by an interviewer. The survey included questions about sociodemographic characteristics, sexual behavior, and sexual partner network, and history of HIV testing and counseling. After the interview, participants were tested for HIV, free of charge. All questionnaires were completed at the Kaihua, Jiangshan, or Lucheng Center for Disease Control (CDC).

## HIV testing

A venous blood specimen (5 mL) was drawn from each participant following the interview. The HIV screening test was performed using an enzyme-linked immunoassay, and if the result was positive, the specimen was retested using another enzyme-linked immunosorbent assay (ELISA) kit. If the results of one or both ELISA tests were positive, a western blot assay was conducted to confirm the diagnosis.

## Statistical analysis

Data were entered into EpiData version 3.1 (http://www.epidata.dk/) via double entry. Once the data had been cleaned and verified, statistical analysis was performed. For descriptive analyses, categorical variables were presented as frequencies and proportions, while continuous variables were presented as medians and interquartile ranges (IQRs), or means and standard deviations (SDs). The significance of difference in the general demographic characteristics was tested using a chi-square ($\chi^2$) tests. Factors from the univariate analysis with $P<0.10$ and/or those previously shown to be associated with the differences in sociodemographic characteristics among rural and urban MSM were included in multivariate logistic regression models, and AORs were calculated along with coinciding 95% confidence intervals (CIs). The statistical significance level was $P \leq 0.05$ and $\beta = 0.1$. Data were analyzed using SPSS version 19.0 (IBM Corp, Armonk, NY, USA).

## Ethics statement

The study was approved by the Zhejiang Provincial Center for Disease Control and Prevention. Written informed consent was signed by all participants during the survey. Participants received 100 RMB (approximately $16 US), free condoms, lubricants, and health counseling for their participation in the survey. Participants with positive results for the HIV test were informed and counseled by the staff of the Kaihua, Jiangshan or Lucheng CDC and received

the necessary referral services. All the study procedures were carried out in accordance with the approved guidelines and regulations.

## Results

A total of 710 MSM participated in the study, 256 (36.1%) lived in rural areas and 454 (63.9%) lived in urban areas. Participants had a median age of 32 years (IQR: 26–40 years), with approximately two-thirds of participants being under the age of 35 years, 44.9% were married or cohabitating, and 79.9% had health insurance. Over half the participants had found sexual partners through the dating apps (software for MSM such as blued, jack'd).

Participants' sociodemographic characteristics are shown in Table 1. Of the 710 participants, 118 (16.6%) participants were diagnosed as HIV positive. Participants' median age at the first homosexual sexual act was 23 years (IQR; 20–29 years). Regarding sexual risk behaviors, 18.0% of the participants reported having had >3 male sexual partners, 61.1% reported not using condoms with their male sexual partner(s) in the past 6 months, and 25.9% reported had not using condoms with their female sexual partner(s). In addition, 45.8% had not accepted an HIV intervention in the past year, and 57.0% had not been tested for HIV in the past year.

There were significant associations between area of residence and age, marital status, level of education, health insurance, sexual orientation, the age at the first homosexual sex act, sexual behavior, male sexual partner condom use, the possibility of being infected with HIV and having been diagnosed with HIV.

Among participants living in rural areas, 17.6% reported that they had had sex with men living in cities in Zhejiang Province, and 10.9% reported that they had had sex with men in the other provinces (Table 2).

The results of the multivariate logistic regression analysis comparing MSM in urban and rural areas are shown in Table 3. Compared to the participants living in rural areas, the participants living in urban areas were more likely to be married or cohabiting with a woman showed the factors of being female married or cohabitation(AOR: 2.32, 95% CI: 1.33–4.05), have a low level of education time (AOR: 2.39, 95% CI: 1.28–4.45), not have local health insurance (AOR: 21.65, 95% CI: 6.28–74.68), through friend introduction or place found sexual partner (AOR: 6.08, 95% CI: 3.70–9.97), practice exclusively homosexual behavior (AOR: 3.22, 95% CI: 1.90–5.45), have used condoms in the past 6 months (AOR: 7.46, 95% CI: 4.45–12.52), believing that they were not at risk of HIV infection(AOR: 2.99, 95% CI: 1.58–5.82) and being HIV positive (AOR: 4.48, 95% CI: 1.91–10.47).

Of the participants, 43.0% had undergone HIV testing in the past year. Multivariate logistic regression analysis showed that HIV testing in the past year was associated with living in urban area (AOR: 1.81, 95% CI: 1.18–2.79), being single or divorced (AOR: 1.59, 95% CI: 1.10–2.30), having a monthly income of at least 4000 yuan (AOR: 2.04, 95% CI 1.19–3.51), having urban health insurance (AOR: 1.97, 95% CI: 1.32–2.95), having found sexual partner using dating apps(AOR: 2.22, 95% CI: 1.51–3.26) and have accepted HIV intervention in the past year (AOR: 6.78, 95% CI: 4.73–9.72(Table 4).

## Discussion

This study is the first on of HIV infection, sexual behavior, HIV testing and associated factors among MSM population in rural and urban areas of living in Zhejiang Province. The HIV prevalence was 16.6% overall (3.9% among those living in rural areas and 24.2% among those living in urban areas). In comparison, other studies conducted during the same time period found a prevalence of HIV infection of approximately 8.0% among MSM in urban areas[1, 16,

**Table 1.  Social demographic and behavioral characteristics among rural and urban men who have sex with men.**

| Variables | Total | Rural | Urban | $\chi^2$ | P |
|---|---|---|---|---|---|
| **Total** | 710 (100) | 256 (100) | 454 (100) | | |
| **Age Y IQR 32(26–40)** | | | | 19.607 | <0.001 |
| 16–24 | 166 (23.4) | 75 (29.3) | 91 (20.0) | | |
| 25–29 | 146 (20.6) | 62 (24.2) | 84 (18.5) | | |
| 30–34 | 132 (18.6) | 49 (19.1) | 83 (18.3) | | |
| ≥35 | 266 (37.5) | 70 (27.3) | 196 (43.2) | | |
| **Marital status** | | | | 7.455 | 0.024 |
| Female married/cohabitation | 263(37.0) | 79 (30.9) | 184 (40.5) | | |
| Male cohabitation | 56(7.9) | 19 (7.4) | 37 (8.1) | | |
| Single/divorced | 391(55.1) | 158 (61.7) | 233 (51.3) | | |
| **Education** | | | | 90.887 | <0.001 |
| Junior high school and below | 335 (47.2) | 63 (24.6) | 272 (59.9) | | |
| High school and junior | 221 (31.1) | 100 (39.1) | 121 (26.7) | | |
| college | 154 (21.7) | 93 (36.3) | 61 (13.4) | | |
| **Monthly income yuan** | | | | 3.810 | 0.149 |
| <2000 | 112 (15.8) | 34 (13.3) | 78 (17.2) | | |
| 2000–3999 | 390 (54.9) | 137 (53.5) | 253 (55.7) | | |
| ≥4000 | 208 (29.3) | 85 (33.2) | 123 (27.1) | | |
| **Health insurance** | | | | 140.655 | <0.001 |
| Rural | 328 (46.2) | 106 (41.4) | 222 (48.9) | | |
| Urban | 239 (33.7) | 147 (57.4) | 92 (20.3) | | |
| None | 143 (20.1) | 3 (1.2) | 140 (30.8) | | |
| **Sexual orientation** | | | | 16.763 | <0.001 |
| Homosexual | 327 (46.1) | 143 (55.9) | 184 (40.5) | | |
| Bisexual | 357 (50.3) | 108 (42.2) | 249 (54.8) | | |
| Uncertain | 26 (3.7) | 5 (2.0) | 21 (4.6) | | |
| **Path found sexual partner** | | | | 122.418 | <0.001 |
| Dating apps | 384 (54.1) | 209 (81.6) | 175 (38.5) | | |
| Friend introduction/place | 326 (45.9) | 47 (18.4) | 279 (61.5) | | |
| **First homosexual age Y IQR 23(20–29** | | | | 16.515 | <0.001 |
| <18 | 62 (8.7) | 28 (10.9) | 34 (7.5) | | |
| 18–24 | 350 (49.3) | 146 (57.0) | 204 (44.9) | | |
| ≥25 | 298 (42.0) | 82 (32.0) | 216 (47.6) | | |
| **Sexual behavior** | | | | 18.716 | <0.001 |
| Homosexual | 406 (57.2) | 119 (46.5) | 287 (63.2) | | |
| Bisexual | 304 (42.8) | 137 (53.5) | 167 (36.8) | | |
| **Number of male sexual partner in the past year** | | | | 3.352 | 0.340 |
| 1 | 277 (39.0) | 97 (37.9) | 180 (39.6) | | |
| 2 | 183 (25.8) | 71 (27.7) | 112 (24.7) | | |
| 3 | 122 (17.2) | 49 (19.1) | 73 (16.1) | | |
| ≥4 | 128 (18.0) | 39 (15.2) | 89 (19.6) | | |
| **Male sexual condom use in the past 6 months** | | | | 110.349 | <0.001 |
| Yes/not happen | 276 (38.9) | 34 (13.3) | 242 (53.3) | | |
| No | 434 (61.1) | 222 (86.7) | 212 (46.7) | | |
| **Female sexual condom use in the past 6 months** | | | | 0.425 | 0.514 |
| Yes/not happen | 526 (74.1) | 186 (72.7) | 340 (74.9) | | |
| No | 184 (25.9) | 70 (27.3) | 114 (25.1) | | |

*(Continued)*

**Table 1.** (Continued)

| Variables | Total | Rural | Urban | $\chi^2$ | P |
|---|---|---|---|---|---|
| **Self-assessment the possibility of being infected with HIV** | | | | 67.744 | <0.001 |
| Probably | 131 (18.5) | 59 (23.0) | 72 (15.9) | | |
| Unlikely | 332 (46.8) | 158 (61.7) | 174 (38.3) | | |
| Impossible | 247 (34.8) | 39 (15.2) | 208 (45.8) | | |
| **Ever accepted HIV intervention in the past year** | | | | 1.504 | 0.220 |
| No | 325 (45.8) | 125 (48.8) | 200 (44.1) | | |
| Yes | 385 (54.2) | 131 (51.2) | 254 (55.9) | | |
| **Ever HIV testing in the past year** | | | | 0.220 | 0.639 |
| No | 405 (57.0) | 149 (58.2) | 256 (56.4) | | |
| Yes | 305 (43.0) | 107 (41.8) | 198 (43.6) | | |
| **HIV diagnose** | | | | 46.696 | <0.001 |
| Negative | 592 (83.4) | 246 (96.1) | 346 (76.2) | | |
| Positive | 118 (16.6) | 10 (3.9) | 108 (23.8) | | |

18]. Another RDS study found an HIV infection of 13.8% in MSM living in urban areas in Zhejiang Province[19], and another study found an HIV prevalence of 3.6% in MSM living in rural areas in the Hebei Province[20], which is consistent with the results of our study. The study conducted in Hebei Province indicated that HIV infection had been spread among the rural MSM population, while the present study found that the prevalence of high-risk sexual behavior, low condom use, low HIV testing and urban-rural mobility was associated with HIV infection among MSM living in rural areas. The high HIV prevalence, high prevalence of high-risk behavior, and low HIV testing rate common in rural areas suggests that in order to achieve the goal of diagnosing 90% of HIV infections, immediate interventions are needed.

The present study found that 43.0% of the participants had been tested for HIV in the past year, which was lower than the 55.9% and 56.8% reported in studies of MSM living in the urban centers of Zhejiang Province[19] and Hangzhou City[1], respectively, and lower than the 2009 national HIV testing rate of 51.2%[21] and lower than the 48.1% HIV testing rate reported from an online recruitment study of MSM in China[22]. The percentage was also lower than the reported HIV testing rates of 69%[4] and 58.2%[5] in Beijing and Shenzhen, respectively. The HIV testing rate was far below the rate reported in many developed countries, such as 67% in the United States in 2011[23] and 72.4% in Australia in 2014[24]. Specifically, in a study of MSM conducted in the United States, approximately half of participants had been tested for HIV within the past 30 days[25]. However, the rate of the present study was much higher than a study conducted in South Africa that reported that 17.5% of

**Table 2. The homosexual behavior among migrant rural MSM in the past year.**

| Variables | N (%) | Sexual behavior | | $\chi^2$ | P |
|---|---|---|---|---|---|
| | | Bisexual | Homosexual | | |
| **Had homosexual in cities of Zhejiang in the past year** | | | | 0.470 | 0.493 |
| No | 211(82.4) | 115(83.9) | 96(80.7) | | |
| Yes | 45(17.6) | 22(16.1) | 23(19.3) | | |
| **Had homosexual in other provinces in the past year** | | | | 0.635 | 0.426 |
| No | 228(89.1) | 124(90.5) | 104(87.4) | | |
| Yes | 28(10.9) | 13(9.5) | 15(12.6) | | |

**Table 3. The difference of social demographic and behavioral characteristics among rural and urban men who have sex with men.**

| Variables | Urban % (n/N) | OR 95% CI | AOR 95% CI |
|---|---|---|---|
| **Overall** | 63.9 (454/710) | | |
| **Marital status** | | | |
| Female married/cohabitation | 70.0 (184/263) | 1.58 (1.13–2.20) | 2.32 (1.33–4.05) |
| Male cohabitation | 66.1 (37/56) | 1.32 (0.73–2.38) | 1.17 (0.51–2.68) |
| Single/divorced | 59.6 (233/391) | 1.00 | 1.00 |
| **Education** | | | |
| Junior high school and below | 81.2 (272/335) | 6.58 (4.31–10.05) | 2.39 (1.28–4.45) |
| High school and junior | 54.8 (121/221) | 1.85 (1.22–2.80) | 1.57 (0.88–2.79) |
| college | 39.6 (61/154) | 1.00 | 1.00 |
| **Health insurance in local** | | | |
| Rural | 67.7 (222/328) | 1.00 | 1.00 |
| Urban | 38.5 (92/239) | 0.30 (0.21–0.42) | 0.38 (0.24–0.60) |
| None | 97.9 (140/143) | 22.28 (6.93–71.57) | 21.65 (6.28–74.68) |
| **Path found sexual partner** | | | |
| Dating apps | 45.6 (175/384) | 1.00 | 1.00 |
| Friend introduction/place | 85.6 (279/326) | 7.09 (4.90–10.25) | 6.08 (3.70–9.97) |
| **Sexual behavior** | | | |
| Homosexual | 70.7 (287/406) | 1.98 (1.45–2.70) | 3.22 (1.90–5.45) |
| Bisexual | 54.9 (167/304) | 1.00 | 1.00 |
| **Male sexual condom use in the past 6 months** | | | |
| Yes/not happen | 87.7 (242/276) | 7.45 (4.97–11.18) | 7.46 (4.45–12.52) |
| No | 48.8 (212/434) | 1.00 | 1.00 |
| **Self-assessment the possibility of being infected with HIV** | | | |
| Probably | 55.0 (72/131) | 1.00 | 1.00 |
| Unlikely | 52.4 (174/332) | 0.90 (0.60–1.35) | 1.11 (0.62–2.00) |
| Impossible | 84.2 (208/247) | 4.37 (2.69–7.10) | 2.99 (1.58–5.82) |
| **HIV diagnose** | | | |
| Negative | 58.4 (346/592) | 1.00 | 1.00 |
| Positive | 91.5 (108/118) | 7.69 (3.94–14.98) | 4.48 (1.91–10.47) |

participants had undergone recent HIV testing[26]. Our results showed that rural areas had a lower HIV testing rate than urban areas, thus, the present study suggests that more measures should be implemented among rural MSM populations to achieve 90% infections diagnosed by 2020.

Our study showed that 81.6% of MSM living in rural areas found sexual partners through dating apps. Studies conducted in developed countries have similarly reported the internet to be the most popular method among MSM for meeting sexual partners[27]. Rural areas are geographically isolated from centers of gay culture and tend to have limited public meeting spaces for MSM to gather and meet potential sexual partners[28]. Dating apps are an ideal method of meeting sexual partners in rural areas because users can remain anonymous and the cost of traveling long distances to urban areas can be eliminated[29]. Several studies have also found that dating apps and the internet may offer new HIV prevention opportunities for MSM living in rural areas[30–35]. The dating apps provides a useful and low-cost approach to recruiting and assessing HIV sexual risks for MSM living in rural areas. Further, studies have

**Table 4. Factors associated with HIV testing in the past year among rural and urban men who have sex with men.**

| Variable | HIV Testing % (n/N) | OR 95% CI | AOR 95% CI |
|---|---|---|---|
| **Overall** | 43.0 (305/710) | | |
| **Aera** | | | |
| Rural | 41.8 (107/256) | 1.00 | 1.00 |
| Urban | 43.6 (198/454) | 1.08 (0.79–1.47) | 1.81 (1.18–2.79) |
| **Marital status** | | | |
| Female married/cohabitation | 34.2 (90/263) | 1.00 | 1.00 |
| Male cohabitation | 46.4 (26/56) | 1.67 (0.93–2.99) | 1.33 (0.70–2.55) |
| Single/divorced | 48.3 (189/391) | 1.80 (1.30–2.48) | 1.59 (1.10–2.30) |
| **Monthly income yuan** | | | |
| <2000 | 36.6 (41/112) | 1.00 | 1.00 |
| 2000–3999 | 40.5 (158/390) | 1.18 (0.76–1.82) | 1.29 (0.79–2.11) |
| ≥4000 | 51.0 (106/208) | 1.80 (1.12–2.88) | 2.04 (1.19–3.51) |
| **Health insurance** | | | |
| Rural | 38.1 (125/328) | 1.00 | 1.00 |
| Urban | 48.5 (116/239) | 1.53 (1.09–2.15) | 1.97 (1.32–2.95) |
| None | 44.8 (64/143) | 1.32 (0.88–1.96) | 1.34 (0.84–2.15) |
| **Path found sexual partner** | | | |
| Dating apps | 48.4 (186/384) | 1.63 (1.21–2.21) | 2.22 (1.51–3.26) |
| Friend introduction | 36.5 (119/326) | 1.00 | 1.00 |
| **Ever accepted HIV intervention in the past year** | | | |
| No | 21.2 (69/325) | 1.00 | 1.00 |
| Yes | 61.3 (236/385) | 5.88 (4.20–8.22) | 6.78 (4.73–9.72) |

shown that MSM living in rural areas who use the dating apps to meet sexual partners are more likely to engage in high-risk behavior, than those who do not use dating apps[30]. A study conducted in China[36] indicated that HIV testing-related social media websites, such as WeChat, have been used to promotion of HIV testing. Therefore, the dating apps should be considered as a potential mechanism of promoting HIV testing and implementing web-based HIV risk reduction interventions for MSM living in rural areas.

Consistent with previous studies, we found that participants who had accepted interventions in the past year were more likely to promote future HIV testing. A study conducted in Zhejiang[19] reported that MSM who had received an AIDS/STIs educational intervention in the past year had a higher rate of HIV testing than those who did not receive the intervention. This indicates that HIV interventions should be made more widely available. However, of the MSM who had accepted an intervention, only 61.3% had undergone HIV testing, suggesting the need for additional studies to determine why MSM are reluctant to be tested for HIV. Efforts should be made to increase HIV testing uptake among MSM population living in rural areas.

Our study found that nearly one-third of participants were married to or cohabitating with a woman and 25.9% had not used a condom during heterosexual intercourse within the past year, which is consistent with previous studies[37]. The low use of condoms by MSM in heterosexual relationships increases the risk of transmitting HIV[37]. Similar to the results of our study, a study conducted in Shenzhen revealed that HIV risk behaviors were more common among MSM engaged in heterosexual relationships as opposed to MSM who engaged solely in homosexual and heterosexual relationships than in MSM who engaged solely in homosexual relationships [38]. Our study indicates that MSM who engage in homosexual behavior in

rural-urban areas could potentially act as a bridge for HIV transmission to the general female population; thus, more attention should be paid to high-risk behaviors among MSM who are sexually involved with both men and women.

We found the rural MSM had reported high risk behavior than urban MSM, yet paradoxically had a lower rate of HIV infection than unban MSM. In China, especially in rural areas, cultural and familial pressures tend to make MSM unwilling to disclosure their sexual orientation. Thus, MSM tend to prefer to gather in the urban areas which are more inclusive and accepting of MSM. The HIV epidemic among MSM in China is concentrated mainly in urban cities[2, 19]. Our study showed that some MSM living in rural areas had engaged in sex with men in cities during the previous year, and that some unban MSM had health insurance in rural areas. The above results indicated that rural MSM had a lower prevalence of HIV infection because the HIV epidemic had longer time in urban areas, resulted in urbans areas had more HIV-infected MSM and high HIV prevalence, while the HIV epidemic had shorter time in rural areas, had little HIV-infected MSM and low HIV prevalence. However, with the higher high-risk behavior and more HIV-infected MSM rural areas, there is a potential for a rapid increase in the incidence of HIV infection in the rural population. Additional intervention measures should immediately be implemented among rural MSM population to reduce HIV transmission.

A study in Vietnam found that MSM living in rural areas had less access to specific HIV prevention information on homosexual sex and had less knowledge on how to protect themselves from HIV infection[39]. In contrast to the previous study, our study found that at least 17.6% of the MSM in rural had sexual behavior in the cities, with migrant MSM potentially transmitting HIV to rural areas. In another study of MSM conducted in China[40], the rate of high-risk behaviors among migrants was 5.8%, which was higher than the 2.8% for rural residents and the 1.0% for urban residents. HIV intervention methods should be targeted at rural-to-urban MSM.

Although RDS is a method that approximates probability sampling of a hidden population, the present study showed that using RDS can recruit more MSM participants living in rural areas, as compared to the routine sampling, and can provide an effective way to determine HIV prevalence and testing rates in rural and urban areas. However, there were several limitations in our study. First, the sample size was inadequate in rural areas, even though the survey recruited more than two times the number of MSM than the number recruited in previous studies. Several important potential associations could not be properly assessed because of the limited sample size. Second, the study was costly due to the use of financial incentives to recruit participants. It may not be feasible to use RDS methods to recruit MSM in future studies because of the high cost involved. Finally, the main purpose of the original survey was to determine HIV testing rates in the previous year and to examine factors associated with HIV testing; However, the study did not consider certain factors known to be associated with HIV testing, such as social support and stigmatization.

## Conclusion

Our study found that high-risk behaviors were common and HIV testing rates were low among MSM living in rural areas. Additional intervention measures should be implemented among rural MSM urgently to reduce HIV transmission and to promote HIV testing.

## Supporting information

**S1 File.**
(DOCX)

**S2 File.**

(DOCX)

## Acknowledgments

Thanks are expressed to the participants for their contributions to the study.

## Author Contributions

**Conceptualization:** Lin He.

**Data curation:** Lin He, Jun Jiang.

**Formal analysis:** Lin He.

**Funding acquisition:** Xiaohong Pan.

**Investigation:** Lin He, Jiezhe Yang, Qiaoqin Ma, Jun Jiang, Wei Wang, Jiaquan Qiu, Yazhou Zou, Ping Wang, Dongshe Zhao, Hui Wang, Tingting Jiang.

**Methodology:** Lin He, Xiaohong Pan, Qiaoqin Ma.

**Resources:** Lin He, Jiezhe Yang, Qiaoqin Ma.

**Software:** Lin He.

**Supervision:** Lin He.

**Writing – original draft:** Lin He.

**Writing – review & editing:** Lin He, Xiaohong Pan.

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
