## [Decision Letter · Decision Letter 0]

14 Jan 2020

PONE-D-19-33057

Higher high-risk sexual behavior and lower HIV testing in rural areas men who have sex with men in Zhejiang province, China: a respondent-driven sampling study

PLOS ONE

Dear Mrs Pan,

Thank you for submitting your manuscript to PLOS ONE. After careful consideration, we feel that it has merit but does not fully meet PLOS ONE’s publication criteria as it currently stands. Therefore, we invite you to submit a revised version of the manuscript that addresses the points raised during the review process.

1.Author needs to provide the sampling principle, and why choosed this rural district and urban area for study?

2.How about the overall epidemic condition of MSM in Zhejiang province and China? I can not find the necessarity of this investigation.

3.The writing needs to be improved substantially.

4.Just simple statistical analysis was performed and the research method did not support the conclusion effectively.

We would appreciate receiving your revised manuscript by Feb 28 2020 11:59PM. To enhance the reproducibility of your results, we recommend that if applicable you deposit your laboratory protocols in protocols.io, where a protocol can be assigned its own identifier (DOI) such that it can be cited independently in the future. For instructions see: http://journals.plos.org/plosone/s/submission-guidelines#loc-laboratory-protocols

We look forward to receiving your revised manuscript.

Kind regards,

Zhefeng Meng, M.D., Ph.D.

Academic Editor

PLOS ONE

Journal Requirements:

2. Please address the following:

- Please include additional information regarding the survey or questionnaire used in the study and ensure that you have provided sufficient details that others could replicate the analyses. For instance, if you developed a questionnaire as part of this study and it is not under a copyright more restrictive than CC-BY, please include a copy, in both the original language and English, as Supporting Information. In addition, please describe any pre-testing of the questionnaire that was performed, including how many participants were involved and where they were recruited from.

- Please ensure you have thoroughly discussed any potential limitations of this study within the Discussion section, including the potential biases introduced by the sampling and data collection methods.

- Please refer to any sample size calculations performed prior to participant recruitment. If these were not performed please justify the reasons. Please refer to our statistical reporting guidelines for assistance (https://journals.plos.org/plosone/s/submission-guidelines.#loc-statistical-reporting).

Reviewers' comments:

Reviewer's Responses to Questions

**Comments to the Author**

1. Is the manuscript technically sound, and do the data support the conclusions?

Reviewer #1: Yes

2. Has the statistical analysis been performed appropriately and rigorously? 

Reviewer #1: Yes

3. Have the authors made all data underlying the findings in their manuscript fully available?

Reviewer #1: No

4. Is the manuscript presented in an intelligible fashion and written in standard English?

Reviewer #1: No

5. Review Comments to the Author

Reviewer #1: The manuscript is focusing on the differences between rural and unban MSMs in HIV testing and infection, ect. Despite of higher proportion of high risk behavior, there was a lower rate of HIV infection in rural MSMs, as compared to unban MSMs. The authors did not analyze the reasons for the paradox. From the description, a proportion of the study rural MSMs had homosexual behavior in cities during last year, and some unban MSMs had health insurance in rural areas. Are the two study population discrete or mixed up?

6. PLOS authors have the option to publish the peer review history of their article (what does this mean?). If published, this will include your full peer review and any attached files.

Reviewer #1: No

---

## [Author Response · Author response to Decision Letter 0]

27 Feb 2020

Dear Editor:

On behalf of my co-authors, we thank you very much for giving us an opportunity to revise our manuscript, we appreciate editor and reviewers very much for their positive and constructive comments and suggestions on our manuscript entitled “HIV risk behavior and HIV testing among rural and urban men who have sex with men in Zhejiang Province, China: a respondent-driven sampling study”. (PONE-D-19-33057). We have studied reviewer’s comments carefully and have made revision which marked in red in the paper. We have tried our best to revise our manuscript according to the comments. Attached please find the revised version, which we would like to submit for your kind consideration.

We would like to express our great appreciation to you and reviewers for comments on our paper. Looking forward to hearing from you.

Thank you and best regards.

Yours sincerely,

Lin He

Corresponding author:

Xiaohong Pan

---

## [Editor Report · Decision Letter 1]

16 Mar 2020

HIV risk behavior and HIV testing among rural and urban men who have sex with men in Zhejiang Province, China: a respondent-driven sampling study

PONE-D-19-33057R1

Dear Dr. Pan,

We are pleased to inform you that your manuscript has been judged scientifically suitable for publication and will be formally accepted for publication once it complies with all outstanding technical requirements.

With kind regards,

Zhefeng Meng, M.D., Ph.D.

Academic Editor

PLOS ONE
---

## [Editor Report · Acceptance letter]

18 Mar 2020

PONE-D-19-33057R1 

HIV risk behavior and HIV testing among rural and urban men who have sex with men in Zhejiang Province, China: a respondent-driven sampling study 

Dear Dr. Pan:

I am pleased to inform you that your manuscript has been deemed suitable for publication in PLOS ONE. Congratulations! Your manuscript is now with our production department. 

With kind regards,

on behalf of

Dr. Zhefeng Meng 

Academic Editor

PLOS ONE